# Current Approaches to Salvage Surgery for Head and Neck Cancer: A Comprehensive Review

**DOI:** 10.3390/cancers15092625

**Published:** 2023-05-05

**Authors:** Romina Mastronicola, Pauline Le Roux, Aurore Casse, Sophie Cortese, Emilie Beulque, Marco Perna, Gilles Dolivet

**Affiliations:** 1Institut de Cancérologie de Lorraine ICL, 6 Avenue de Bourgogne, 54519 Vandoeuvre-lès-Nancy, Francegdolivet@nancy.unicancer.fr (G.D.); 2CRAN, CNRS, UMR 7039, Université de Lorraine, 54519 Vandoeuvre-lès-Nancy, France; 3Technoport 9, Avenue des Hauts-Fourneaux, 4362 Esch-sur-Alzette, Luxembourg

**Keywords:** head and neck cancer, salvage surgery, free flap, transoral robotic surgery, navigation surgery, sentinel node mapping

## Abstract

**Simple Summary:**

As of today, salvage surgery regarding head and neck cancer remains a challenge for patients, surgeons, and oncologists. Indeed, even though several advances have been made in the last few years, the results in terms of both survival and complications are disappointing. Therefore, it remains important to be aware of the latest emerging techniques and understand their limits to make progress in this field. In addition, the patient’s conditions play a major role in the outcome of salvage surgery, so identifying the various factors influencing the results can help improve these approaches. This review will walk us through the latest literature on salvage surgery, and it will help us understand the several difficulties touching this last treatment resort.

**Abstract:**

Salvage surgeries of head and neck cancer are often complicated and do not always show decent results. This type of procedure is tough on the patient, as many crucial organs can be affected. A long period of reeducation usually follows the surgery because of the need to rehabilitate functions such as speech or swallowing. In order to lighten the journey of the patients, it is important to develop new technologies and techniques to ease the surgery and limit its damages. This seems even more crucial since progress has been made in the past years, allowing more salvage therapy to take place. This article aims at showing the available tools and procedures for salvage surgeries, such as transoral robotic surgery, free-flap surgery, sentinel node mapping, and many others, that help the work of the medical team to operate or obtain a better understanding of the status of the cancer when taken in charge. Yet, the surgical procedure is not the only thing determining the outcome of the operation. The patient themself and their cancer history also play an important part in the care and must be acknowledged.

## 1. Introduction

In 2020, head and neck cancer ranked as one of the ten most prevalent types of cancer worldwide [1]. Men are most likely to be concerned by this cancer. Nevertheless, in the last decade, we can observe an increasing incidence for women [2]. Three well-known factors have been identified as accountable for HNC: alcohol, tobacco, and human papillomavirus (HPV) infections, the last one increasing due to the decline of alcohol and tobacco consumption and changing sexual practices over time [3]. Head and neck squamous cell carcinoma can develop in various locations, including the oral cavity, oropharynx, hypopharynx, larynx, and nasopharynx. In the last ten years, there has been a change in the distribution of the primary cancer sites, with a gradual rise in the occurrence of oropharyngeal squamous cell carcinoma and a decrease in cases of cancers affecting the larynx and hypopharynx [4]. Based on the stage of the disease, the treatment will be adapted. It is usually a complex treatment, with specific guidelines that have changed over time. For stage I–II tumors, monotherapies such as surgery or radiotherapy are typically employed, whereas combination treatments are used for the more advanced stage III and IV tumors [5]. Nowadays, the role of salvage surgery is increasing, as the initial treatment (usually radiotherapy) fails in about 50% [6]. Salvage surgery is usually the best treatment option in recurrent locoregional head and neck cancer, as new irradiation is impossible most of the time. It needs to be considered with a resection technique, assuring the total excision of the disease, regardless of the physical and functional consequences. Therefore, operable patients with resectable tumors must face radical surgery in addition to cancer relapse. Surgical techniques have evolved to preserve anatomical structures by using less invasive approaches by limiting the surgical procedure by providing neoadjuvant treatments, or by limiting the surgical excision itself without compromising the oncological result and the survival of the patient.

However, the results concerning survival and complications are not convincing, especially in advanced stages. These prognoses can be explained by several clinical and pathological factors. Some of these factors include the initial and recurrent stage of the condition, the status of the resection margins, the anatomical location of the condition, HPV status, and the time elapsed between the initial treatment and salvage surgery [7]. Therefore, it remains important to elaborate new strategies to improve surgery outcomes, and to identify all the factors that play a role in failures.

The aim of this review is to present the latest literature on salvage surgery in head and neck cancer, allowing for a better understanding of this treatment.

## 2. Head and Neck Cancer Resections

Radical neck dissection was first performed in 1951. It included the removal of all the lymph nodes between the mandible and the clavicle, with an incision under the lower lip and segmental mandibulectomy and in-continuity radical neck dissection.

Usually during the first operation, lymphatic structures are removed in order to limit the risks of cancer recidivism. In the modified radical neck dissection, only the lymph nodes from group I to V are removed.

The neck dissection is realized in the form of a U or Y shape, which can be esthetically unpleasing for the patient. Robotic-assisted surgeries prevent the making of a long scar, which is less traumatizing for the patient [8].

## 3. Transoral Robotic Surgery (TORS)

Transoral robotic surgery is a minimally invasive surgical procedure that uses a robotic system to perform surgery through the mouth. The development of TORS in head and neck cancers has enabled the excision of large tumors, previously treated by very mutilating and dangerous surgeries. Indeed, robot arms and articulation allow easy access to the tumor. Robotic surgery is more precise with 3D vision; thus, we can explore and excise unusual and difficult parts to reach (Figure 1). Compared to more traditional surgical approaches, TORS offers recurrent tumors better outcomes for selected patients. In a study by White et al. [9], we noticed shorter operative times and hospital stays, and less postoperative complications and functional disability, such as speech or swallowing functions. Several tools are in development and have already demonstrated convincing results for this surgical technique, such as the Flex Robotic System potentially replacing the DaVinci Si HD system already in place [10].

Dabas et al. [11] explain that, in a study conducted on 30 patients who underwent TORS with the DaVinci robot, the positive margins after the excision were only in 6.7% of the patients. The oral feeding could start as early as the third day postoperation, and long-term gastrostomy-tube dependency only occurred in 10% of the studied population. Among the 30 patients who participated in this study, 3 of them passed away, with 1 death attributed to nononcological causes. The overall survival rate for the patients was 86% at 122 months, and the median survival time was 19 months (with a range of from 7 to 122 months). On top of that, in White et al. [9], there was a statistically significant difference in the two-year overall survival rates between the TORS approach and open surgical approach (74% vs. 43%). This shows that such a procedure can be used as a safe, effective, and feasible approach to salvage surgeries, as benefits can be observed.

TORS is still recent in head and neck surgery; therefore, this treatment is reserved for well-selected patients in terms of tumor staging and accessibility. It is a promising method, which could help patients enhance their quality of life and ability to function after treatment. Moreover, it is very useful in salvage surgery, as it allows us to be more precise, so that noble structures (nerves, arteries) will be better preserved. Thanks to nerve detectors or nerve stimulators, such structures can indeed be avoided by the surgeon. More studies to come will allow for the more widespread use of TORS in the future [10,12].

Yet, complications may occur even with TORS. The most frequent ones are hemorrhage (2.4–3.1%), pharyngostoma (2.5%), surgery site infection (2.3%), and pneumopathy (0–7%) [13,14].

A French study was conducted on seven cases of spondylodiscitis following malignant pharyngeal tumors operated on with TORS. Nickerson identified risk factors for postoperative spondylodiscitis (a long operative time, severe blood loss, tissue irradiation, extensive soft-tissue dissection, tissue necrosis, and the creation of a dead space), and some of them were identified in the study group. Five patients had local radiation treatment, and one had a dissection extending to the whole posterior pharyngeal wall [15]. Radiation increases the risk of such complications happening, since it puts the tissues at risk for necrosis and superinfection [16,17]. Furthermore, posterior pharyngeal wall resection by TORS with no reconstruction leaves a possible infection entry point, as it is close to the cervical spine. To prevent this, the prevertebral fascia should be conserved, and flap reconstruction should be considered as an option, as well as low-intensity monopolar dissection [15]. In the end, patients with multiple comorbidities are put at risk with surgery for posterior pharyngeal wall tumors without reconstruction with TORS; thus, at the first signs of infection, spondylodiscitis should be investigated early.

## 4. Types of Reconstruction

Since surgery in head and neck cancer is often mutilating and impacts important functions (speech, breathing…), reconstruction surgery is usually needed. In soft-tissue reconstruction, flaps can be harvested in multiple areas, such as the latissimus dorsi, radial forearm, anterolateral or lateral thigh [8], and many more. In hard-tissue reconstruction, an osteocutaneous flap can also be harvested from many sites of the body, such as the scapula, fibula, or iliac crest [8].

Salvage surgery can include the rehabilitation of a failing free flap as an example.

## 5. Cancer Surgery with Free Flap

In the last three decades, microvascular tissue transfer has been a major evolution in head and neck reconstruction. The use of pedicle flaps marked the beginning of reconstructive surgery with the incorporation of a well-vascularized tissue. With free flaps, surgeons have been able to expand their surgical indications while minimizing negative effects on both the appearance and function aspects, thus increasing the range of surgeries they can practice. In the field of oncology, microvascular tissue transfer is currently considered the most reliable and effective technique for reconstructing lost tissue [18,19,20]. However, several factors need to be accounted for in salvage surgery, such as a history of radiotherapy or chemotherapy in the same area, which can make surgical dissection more complicated. Various studies show that extern radiotherapy should not be a contraindication, as it is not responsible for the failure of the surgery [21,22,23]. Indeed, Jones et al. [24] concluded, in a study conducted on 305 patients, that a history of external radiotherapy was not correlated to an increased risk of vascular thrombosis and, by extension, the failure of the procedure in a salvage surgery. Yoshimoto et al. [18,25] tried to clearly identify risk factors for patients undergoing free-flap surgery. According to the results, it appeared that only intraoperative chemoradiation had a statistically significant impact on the outcome of the free-flap salvage surgery. Regarding previous external radiotherapy or previous cervical surgery, they could not be linked to any necrosis of the flap.

It also seems useful to mention how important the preservation of the blood supplies during any cervical surgery is, regarding venous and arterial blood vessels. Any useless waste of the vessels must be avoided. When it comes to the choice of receiver vessels, it seems better to use vessels that are not in an area with a history of radiation. In addition to this, we usually ensure the venous return with a minimum of two veins. The following of a rigorous procedure allowed us to obtain satisfying results, even in critical cases [19].

On top of this, it is important that this surgery is managed by well-trained team surgeons, with experience and preparation. Indeed, L. Dekerle et al. demonstrate a survival rate of 58.67% at ten years for early-stage patients, in a specialized center, which is higher than in other studies [26].

In this study, conducted at “Institut Cancérologique de Lorraine” [26], 59 patients from 1999 to 2007 were analyzed from the database. These patients underwent surgical resections of head and neck tumors during microsurgery, using free flap for reconstruction. The types of free flaps were forearm, fibula, latissimus dorsi, or pectoral. After the surgery, the global survival rate at 10 years was 38.6%, which is better than what is usually seen, and the flap failure rate was 1.85%, the last figure being lower than in the literature. This shows that the experience of the surgical team plays a part in the outcome of the operation. Factors such as TNM score, age, or even cancer history were also identified as impactful on the survival rate. As an example, patients over 60 years old or with a TNM score over 3 had lower outcomes regarding the surgery. Nevertheless, this study did not consider the quality of life of the patient. Indeed, salvage surgeries in the head and neck area can be mutilating and can deeply affect the life of the one who chooses to go through it.

Despite this impact on the quality of life of patients, free-flap reconstruction surgery in head and neck cancer remains one of the best options, with good survival rates and functional outcomes and few complications. Therefore, in terms of benefits/risks, it is worth using this method given all the results obtained.

## 6. Cancer Surgery with Pedicled Flap

Pedicled flaps are still an important option for the reconstruction of defects in salvage surgeries, even if nowadays free flaps tend to be more popular. The careful evaluation of a patient’s overall health and local anatomy is essential for a surgeon to apply advanced reconstructive techniques with caution. In order to select the most suitable reconstructive solution, valid alternatives must be assessed. Despite the widespread use of free flaps, several reports demonstrate the good reliability of pedicled flaps in terms of functionality [26,27,28]. In a study of Mahieu et al. [23], they compared pedicled flaps and free flaps for the reconstruction of defects in head and neck cancer. Out of the 93 patients, 64 had pedicled-flap reconstructions, and the results showed no significant difference compared to the free flap. Several factors were considered, such as the functional outcome, flap necrosis, complications, and prognosis. Diverse pedicled flaps are described in the literature, such as the musculocutaneous infrahyoid flap (IHMC) [29] or latissimus dorsi myocutaneous flap [30]. The IHMC flap is well adapted for medium-sized defects in the oral cavity or pharynx. Indeed, being thin and supple, this flap’s characteristics enable it to conform to a wide range of anatomical structures. Mirghani et al. [29] detailed the surgical key points in order to reduce the risk of necrosis, and they demonstrated its reliability and value. As for the latissimus dorsi flap, it is usually used for large-defect reconstruction. For example, for women with neck defects who wish to avoid breast deformity resulting from pectoralis-flap reconstruction, the latissimus dorsi pedicled flap offers a particularly useful alternative. In addition, it often results in a superior esthetic outcome due to its excellent skin-matching properties in the head and neck areas [30]. Other types that can be mentioned are the pectoralis major flap, sternocleidomastoid muscle flap, or facial artery musculo-mucosal flap. Here is an example of the use of the pedicled flap for the reconstruction of defects in the neck area, practiced in our center at Institut de Cancérologie de Lorraine (Figure 2).

## 7. Sentinel Node Mapping

The sentinel node is the first lymphatic node to receive lymphatic drainage from the tumor. The purpose is to block the growth of cancer cells by filtering and obstructing their progressions. The healthy state of the sentinel lymph node serves as an indicator for the condition of other lymph nodes and can help medical professionals determine appropriate treatment methods. Therefore, lymphoscintigraphy can help avoid unnecessary neck dissection or node irradiation. However, the risk of false-negative results remains a major limitation in this approach [31]. In the head and neck region, lymph nodes represent 30% of all the lymph nodes in our body [32]. Therefore, although this technique was mainly developed for breast carcinomas, its advance in head and neck melanomas is encouraging and important.

In the review of N. de Rosa et al., the conclusion of several studies was that performing a sentinel node biopsy in the head and neck region is linked with a higher false-negative rate than similar procedures in other areas of the body. However, if the sentinel node biopsy does show positive results, then it is a strong predictor of recurrence [33].

This technique is increasingly used, especially for small tumors (T1/T2; N0). In salvage surgery, the sentinel procedure avoids additional surgery in the neck if the sentinel lymph nodes are free, which reduces the risk of postoperative complications while ensuring satisfactory local control [5,17,32,33]. In the future, it will be interesting to see different techniques combined, such as robotic and sentinel lymph biopsy, both aimed at minimizing the impact of oncological surgery on the head and neck region. This topic is still at an early stage of investigation and more studies are coming, with an interesting development perspective [34]. Furthermore, many tools of detection are in development, such as Tc-99 m tilmanocept or indocyanine green, allowing more precision in sentinel node mapping [32,34].

## 8. ICG-Induced NIR Fluorescence Mapping

Cancer cells left behind after salvage surgery can continue to grow and divide, leading to the formation of new tumors or metastasis, which can further reduce the chances of survival. Therefore, achieving negative margins during salvage surgery is crucial to prevent the recurrence of cancer and improve the chances of survival. In fact, after salvage surgery, the positive margins are around 18–22% [35], and usually the identification of the surgical margins is dependent on the surgeon’s expertise, visual examination, and palpation. It was important to develop new methods allowing for better results in differentiating between normal and malignant tissue tumors. In areas that have previously undergone radiation therapy, distinguishing between tumor and healthy tissues becomes complex due to the delayed and irreversible deterioration of tissues (inflammation, vascular alteration, fibrosis).

The use of fluorescence-guided surgery (FGS) with indocyanine green (ICG) has the potential to facilitate the real-time identification of tumor margins, thereby improving margin clearance rates and progression-free survival [36].

A first trial evoked this possibility [37] by injecting ICG in four patients previously treated by surgery and radiotherapy. The aim was to demonstrate that ICG-based fluorescence can assist in the visualization of tumors and their discrimination from normal tissue in irradiated territory. The first two patients had encouraging results compared to patients 3 and 4 with the weakest fluorescence. This could be explained by the presence of a thick necrotic area or old lichen lesions, but more patients are needed to confirm this. However, with the results obtained, this imaging technique is an important tool for detecting and removing tumors due to its ability to provide real-time feedback and contrast during surgery. Therefore, other studies are needed with larger cohorts to obtain more data to exploit and a better comprehension of failure.

## 9. Detection of Circulating Tumor Cells

The Cell Search System was used at Institut de Cancérologie de Lorraine. It is composed of the Auto-Prep system and the CellTracks Analyzer. The first one is used to detect the circulating tumor cells with immunomagnetic cell enrichment. The second one is a semiautomated fluorescence microscope that is connected to analysis software that is able to display the images of the fluorescent cells on a screen [38]. Another technique to detect circulating tumor cells is RT-PCR. More common, the Reverse Transcriptase Polymerase Chain Reaction remains the most precise technique yet to detect and quantify mRNA. Such techniques are interesting as they allow surgeons to visualize the cancerous cells and to see if their surgery has an impact on the cell release. By doing so, cancer relapses could be prevented, and future salvages surgeries could be limited.

## 10. Salvage-Surgery-Associated Treatments

Numerous studies have shown that, in some cases, salvage surgeries can be combined with other treatments with oncological or rehabilitating aims. Furthermore, when surgery is not an option, the use of radiotherapy technology appears crucial. In tissues that have already been irradiated during previous therapies, different forms of “conformational” irradiation can be used, such as brachytherapy, 3D-conformational radiation therapy (3D-CRT), intensity-modulated radiation (IMRT), and highly conformational radiotherapy.

Brachytherapy is a form of radiotherapy allowing for the delivery of the radiation directly near or inside the tumor. Brachytherapy with a low output dosage represents a good conservative alternative to surgery. A study conducted on twenty-two patients showed that salvage surgery on regional lymphatic metastases for head and neck cancer paired with interstitial brachytherapy was beneficial [39]. Nevertheless, its role remains unclear to scientists and should be reinvestigated.

Regarding 3D-CRT, various studies have shown the possibility of a reirradiation by 3D-conformational techniques with similar acute side effects to the ones seen in the first irradiation, but larger delayed side effects [40].

IMRT was, for a long time, the most commonly indicated modality for salvage re-irradiation regarding head and neck cancer. Nowadays, its use has become more controversial due to the high risk of severe chronic toxicity. Yet, this therapy has been found advantageous for patients with recurrent or secondary head and neck cancer because of its high conformity of the target volume and its ability to avoid the healthy tissue [41].

Stereotactic radiotherapy is a radiation technique using numerous radiation beams in order to closely target the tumor. By doing so, the risks of affecting healthy tissues are lowered. Technologies such as Cyberknife^®^ have proved their efficiency during the last decade. A recent study on 110 patients proved that the use of such a reradiation technique in order to treat recurrent upper aerodigestive tract cancer is coherent [42]: from 2007 to 2019, the global survival rate was 43.8% for patients treated with Cyberknife^®^.

## 11. Prognosticators and Patient Selection in Salvage Surgery

Salvage surgery includes many risks for the patient; consequently, it is important that the patient is aware of what he is incurring. Indeed, salvage surgery is rather uncertain as to recovery and postoperative quality of life. As a result, it is relevant to study prognosticators in salvage surgery, as well as patient selection.

In fact, in a recent study of Locatello et al., which estimated the risks and benefits before salvage surgery in recurrent HNC, it was pointed out that for postsalvage surgery at 5 years, the overall survival was 28.3%, and the disease-specific survival was 38.9% [7]. The aim was to estimate different factors that might play a role in survival and complication prediction in order to help both patients and surgeons to make the right decision regarding salvage surgery. A strong prognosticator is the disease-free interval; indeed, the longer it is, the better the prognosis. In the cohort of 234 patients, the median disease-free interval was 20.5 months, and 126 patients had recurrences within two years. For patients with recurrences within 6 months, the overall survival and disease-specific survival were inferior compared to patients with recurrences after 6 months. It also records complications following the Clavien–Dindo classification, studying CD ≥ III complications. With the help of the WUHNCI, the responsibility of the patient’s general comorbidities in survival and complications was demonstrated.

For example, comorbidities such as uncontrolled diabetes or cardiovascular disease have been implicated in postoperative complications, so these issues have to be considered [43]. Salvage surgery for larynx and hypopharynx cancers is challenging and has only moderate success rates in terms of survival. However, the chances of survival without recurrence are higher if the initial and recurrent tumors are in the larynx, if their size and extent are not larger than T4, if the patient had prior chemotherapy treatment, and if the possibility of dissection during salvage surgery exists. Therefore, identifying these factors can help to select patients who may benefit more from salvage surgery [44].

In addition to this, HPV status needs to be considered. Indeed, patients with an HPV-positive status are more likely to survive. This can be explained by the fact that head and neck cancer induced by HPV usually concerns younger and non-smoker patients (in other words, healthier patients), allowing better prognostics [4].

In conclusion, several conditions are needed for salvage surgery to have favorable outcomes. For instance, a young patient without comorbidities who is HPV-positive and with an early-stage local recurrence in the larynx would be considered as the best conditions in terms of the survival and complication venues. However, there are few candidates like this for salvage surgery, and so a discussion between the surgeons and the patient is important in the decision making. Opting for salvage surgery can significantly enhance the outcomes of head and neck cancer treatment, making it the recommended standard of care for an eligible patient [45].

## 12. Navigation Surgery

With the same purpose of performing a safer surgery while preserving the vessels and nerves, as well as allowing for a good tumor resection, surgical navigation has been developed. Indeed, the precision of surgical outcomes heavily relies on visualization; thus, with navigation surgery, it has been possible to have better visibility, gain more knowledge, and therefore provide enhanced care to patients. At first, mostly neurosurgeons used navigation for their surgeries, but oral and maxillofacial surgeons have also started to use this innovative technology. A study of Gangloff et al., conducted at Institut Cancérologique de Lorraine (ICL), reports the findings of 33 surgeries using navigation [46]. Out of the initial group of 31 patients treated, 27 (81.8%) had a quality of life evaluated as “good”, and 4 had a “fair” quality of life. They showed successful results, with an improved quality of life compared to classical surgical procedures. The steps of navigation are to first collect the morphological data of the patient, and then to transfer them to the navigation device. By integrating patient data with real-time information about the location of surgical instruments, navigation technology enables precise and accurate surgical procedures to be performed (Figure 3).

However, some limits to navigation are known, such as bilateral operation [47]. However, as new research emerges and software continues to be developed and refined, this technological advancement is expected to become an essential tool for craniomaxillofacial surgery procedures, allowing for safe resection margins and better precision.

## 13. Complications of Salvage Surgeries

Since there are no well-accepted guidelines regarding salvage surgeries, the complication rates vary from 23% to 67% in salvage surgeries of the head and neck [6]. The comparison is quite impossible considering the absence of uniformity in the reporting of complications. However, the proposition of classification by Dindo and Clavien [48] is being used more to declare data [49]. As an example, in a salvage total laryngectomy, the overall complication rate was 67.5%, and pharyngocutaneous fistula was the most common complication, with a 28.9% rate [50]. In another study with over 38 patients, 19 had no postoperative complications, 5 of them died of frequent and severe bleeding episodes, 4 had salivary fistulas, and 7 had important pharyngostomas [51]. Pujo et al., in a retrospective study from 2005 to 2013, observed 67.3% of one or more locoregional complications, and 32.7% of one or more general complications [44]. Locoregional complications are the most frequent, with fistulas usually being the most common complications [52]. In a review of Hay et al. [53], TORS complications were analyzed, such as bleeding, dysphagia, aspiration-related infections, or local pain. A total of 35% of the patients had complications related to TORS, and 16% were major complications. In patients with squamous cell carcinoma in the head and neck, the risk of complication was higher in surgery for the tumor with concomitant neck dissection than in neck dissection surgery or tumor surgery alone [6]. Therefore, complications in salvage surgeries are still common, and so it is important to take them into account and treat them the right way, as well as to anticipate them.

## 14. Survival Outcomes

In order to better predict the outcomes of salvage surgery, Tan et al. developed a model based on data available before surgery [6]. Concomitant local or regional failure along with a stage IV tumor were considered independent factors. In patients with none of these factors, the two-year OS rate was 83%. With one of these factors, it was 49%, and with these two factors, the OS rate was 0% [54]. Such an observation suggests that patients with stage IV tumors and local or regional failure should not be considered to undergo salvage surgery. Several studies demonstrate similar results regarding overall survival. Delgado et al. [51] show a five-year OS after salvage surgery of 37.90%, and in Pujo et al. [44], it is 36% at five years. These results are rather stable from one study to another, and it is now important to consider the different factors improving survival.

## 15. The Prospects of Salvage Surgery Development

In the last two decades, there has been a steady rise in the percentage of upper aerodigestive tract cancers that are associated with HPV, particularly those affecting the oropharynx. This finding can be attributed to the decrease in alcohol and tobacco use, as well as to changes in sexual practices. As previously said, being HPV-positive is a favorable prognostic, and research concerning the treatment of these tumors is currently in progress. Indeed, it could be interesting to limit their treatment due to a better response to adjuvant treatment (with a lower dose of radiotherapy, for example).

Surgery occurring in head and neck cancer is often mutilating and has heavy consequences on the patient’s life and capabilities, which is why finding and exploring new techniques and new procedures is so important. There is a need for techniques that allow for not only great healing, but also a minimal impact on the patient. Because of the location of the tumors in head and neck cancer, it is not uncommon for abilities such as phonation or deglutition to be altered by salvage surgeries. A popular effort has been made in this direction with the arrival of minimally invasive and robotic techniques and the search for sentinel nodes, as mentioned earlier in the article.

De-escalation techniques should be evaluated by surgeons, starting by choosing a less invasive surgery and the will to preserve the organ and its function. Recently, the most important example of a de-escalation was the use of neoadjuvant chemotherapy by TPF (Docetaxel, Cisplatin, 5-Fluorouracil) in advanced laryngeal cancer [55]. In cases in which the tumors were reduced by 50% of their volume, the patients would go on with radiotherapy or chemoradiation. Such results allowed the surgeon to avoid practicing a radical surgery, and to preserve part of the larynx [56].

The same way of thinking has been applied to cervical lymph node surgery. Not so long ago, it was common to inflict radical neck dissections in addition to the removal of the tumor regarding squamous cell carcinomas. During such dissections, important anatomical structures would be removed, such as the spinal nerve, the internal jugular vein, or even the carotid artery. Because of the knowledge we now have about the lymphatic drainage of tumors, this surgical procedure has been redefined to cause fewer residual signs, as well as to keep its efficiency [57]. Depending on the case, it is possible that only a selective neck dissection can be performed. When the cancer is at an advanced stage, the radical neck dissection is usually evaluated rather than automatically performed.

Technical novelties such as lymphoscintigraphy have been developed in this direction to allow for a better evaluation of the state of the tumor and its irrigation, thus preventing unnecessary surgical procedures and their consequences [58].

In general, head and neck cancer that is induced by HPV is the most likely to respond to a de-escalation therapy, which explains why many clinical trials are targeting them. In addition to this, this type of head and neck cancer affects younger persons rather than older ones. This is why one of the goals of the many studies has been to reduce the toxicity of the treatments in order to avoid having the patients develop other cancers in the future.

Moreover, advanced visualization techniques have been developed, allowing new opportunities for precision tumor resection. For instance, the use of fluorescence-guided surgery with indocyanine green provides a better outlook of the tumorous tissues. Moreover, the development of transoral robotic surgery could be very helpful in the future of salvage surgery, always in the interest of both patients and surgeons.

## 16. Summary of This Review

Several results are presented in this review, for simplicity you can find in Table 1 a summary of the important information for each technique. This also allows a comparison of the techniques according to their main characteristics and their points of improvement. 

## 17. Conclusions

To conclude, salvage surgery remains a major challenge with several obstacles. Indeed, head and neck cancer is known for having a poor prognosis, with an increasing rate of recurrences. In this respect, salvage surgery is the only good alternative proposed for patients with recurrences; therefore, it is important to offer the best tumor control and survival possible. However, the results are not convincing, and so different techniques are in development, and many studies are trying to improve salvage surgery outcomes. In addition, it is interesting to look at patient selection, as not all patients are good candidates for salvage surgery. Different factors have been identified as favorable for this last treatment resort, and surgeons must take them into account. The future of salvage surgery seems promising, allowing better patient management.

## Figures and Tables

**Figure 1 cancers-15-02625-f001:**
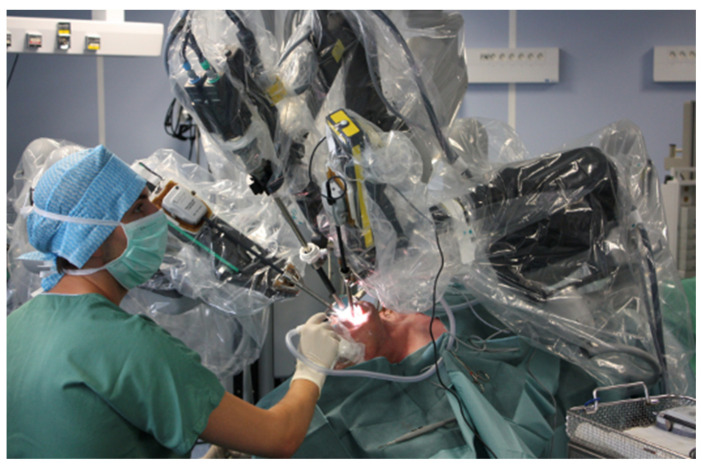
Robot arms introduced transorally during oropharynx surgery.

**Figure 2 cancers-15-02625-f002:**
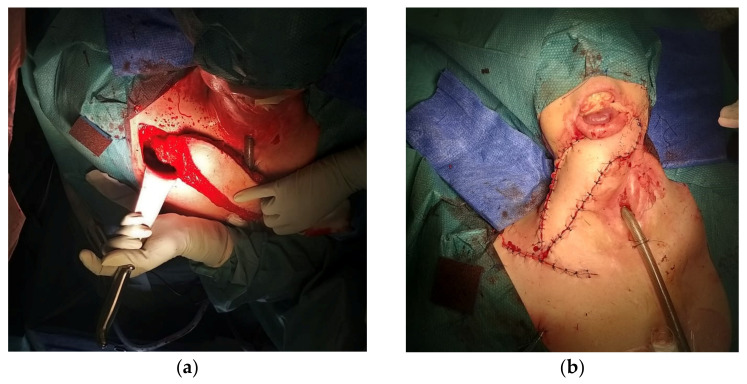
(**a**) During the surgery; (**b**) after the surgery.

**Figure 3 cancers-15-02625-f003:**
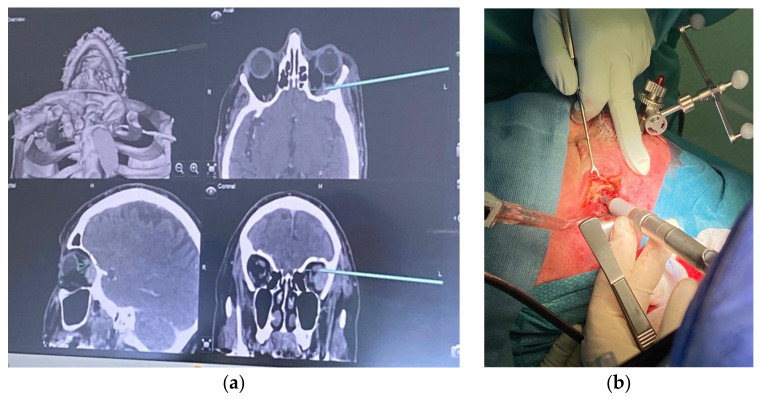
(**a**) Navigation screen with real-time information; (**b**) salvage surgery of periorbital cancer with a navigation device.

**Table 1 cancers-15-02625-t001:** Summary of results presented in this review.

Techniques	Outcomes	Clinical Suggestions
TORS	Shorter operative times and hospital stays Less postoperative complications and functional disability [9] More precise Risk of spondylodiscitis	Applied only to well-selected patients
Cancer surgery with free flap	Good survival rates and functional outcomes Few complications Mutilating	Needs to be performed by well-trained surgeons [26] Extra attention needs to be paid to microvascular complications
Cancer surgery with pedicled flap	Same mortality as free-flap surgery	Improving flap-harvesting techniques
Sentinel node mapping	Strong predictor of recurrence Linked with higher false-negative rate [33]	Many tools in development [45]
ICG-induced NIR fluorescence mapping	Facilitates real-time identification of tumor margins [36]	More studies need to be conducted to better comprehend failure [37]
Navigation surgery	Safer surgery Good tumor resection Improved quality of life [46]	

## Data Availability

No new data were created or analyzed in this study. Data sharing is not applicable to this article.

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
