# Peer review of "Current Approaches to Salvage Surgery for Head and Neck Cancer: A Comprehensive Review"

_cancers, 2023, doi:10.3390/cancers15092625_

Round 1

Reviewer 1 Report

This review reports the last possibility of salvage surgery in clear and simple way, however I am not agree with them , that robotic surgery  " can be used as a safe, effective, and feasible approach to salvage surgeries as benefits can be observed", because actually the survival time is very low (the authors report 19 months of survival time) . Please clarify better what they intent.

Author Response

We appreciate very much your comments to improve our review. Please see the attachment.

Reviewer 2 Report

The manuscript entitled “Salvage Surgery for Head and Neck Cancer Nowadays” has useful information for readers who are interested in this field. I think it could be considered for publication with major revision.

1.       In the section of patient selection, I think you can discuss the relationship between prognosis and the time elapsed between the initial treatment and recurrence with more specific data from previous papers.

2.       I think "chemoradiation" is a more common name than "radio-chemotherapy.

3.       In Line321-322, “the carotid artery or even the internal jugular vein” should be replaced by “the internal jugular vein or even the carotid artery” because it is quite rare to resect the carotid artery.

4.       In Line 324-352, “partial neck dissection” should be replaced by “selective neck dissection”.

5.       In Line 326-327, the authors say “When the cancer is at an advanced stage, the radical neck dissection is usually evaluated rather than automatically done. If it is considered too impactful on the patient, options like radio-chemotherapy seem like a good alternative.” However, I think advanced neck tumors are often difficult to cure with chemoradiation alone, except for HPV-related oropharyngeal cancer. Moreover, since this is a review article in the setting of salvage surgery, chemoradiation is inappropriate.

Author Response

(The authors gave the same response as above.)

Reviewer 3 Report

This a review about salvage surgery in head and neck cancer recurrence.

The authors described some tools that may help the surgeons. Therefore, a more specific title must be chosen.

The aim of the review must be added at the end of the Introduction.

A chapter on pedicled flaps is completely lacking.

The authors shoudl report more data about salvage surgeries (types of resection and reconstruction, complications, survival outcomes).

A table that summarize the results of the review is needed.

Author Response

(The authors gave the same response as above.)

Round 2

Reviewer 2 Report

I think this paper could be considered for publication without further revision.

Author Response

Thank you for your help in improving this review, we really appreciate it. 

Reviewer 3 Report

More data about complications and survival should be added.

The table should not report data from single studies, but must summarize the results of the literature.

Author Response

Thank you for your help in improving this review, we really appreciate it. 

We added more data in paragraphs 13 and 14. 

We created a new table, hoping that it is up to the task. If not, could you be more specific.

Round 3

Reviewer 3 Report

Thank you for improving the manuscript.